# Common transthyretin-derived amyloid fibril structures in patients with hereditary ATTR amyloidosis

Maximilian Steinebrei [1] ✉, Julian Baur[1], Anaviggha Pradhan[1], Niklas Kupfer[1], Sebastian Wiese [2], Ute Hegenbart [3], Stefan O. Schönland [3], Matthias Schmidt [1] & Marcus Fändrich [1]

Systemic ATTR amyloidosis is an increasingly important protein misfolding disease that is provoked by the formation of amyloid fibrils from transthyretin protein. The pathological and clinical disease manifestations and the number of pathogenic mutational changes in transthyretin are highly diverse, raising the question whether the different mutations may lead to different fibril morphologies. Using cryo-electron microscopy, however, we show here that the fibril structure is remarkably similar in patients that are affected by different mutations. Our data suggest that the circumstances under which these fibrils are formed and deposited inside the body - and not only the fibril morphology - are crucial for defining the phenotypic variability in many patients.

For several protein misfolding diseases it was found that the disease manifestations are correlated with the amyloid fibril morphologies that are deposited within the tissue[1,2]. That is, different amyloid fibril morphologies have been found in patients or animals with different pathologies or clinical phenotypes. Examples hereof include the different tau fibril morphologies in different neurodegenerative diseases[3], the different α-synuclein-derived fibrils in Parkinson's disease and multiple system atrophy[4,5], the different Aβ fibril structures in vascular and parenchymal amyloid deposits[6], the different serum amyloid A protein-derived fibrils in the glomerular and vascular variants of systemic AA amyloidosis[7,8] and the different prion protein fibrils in different strains of transmissible spongiform encephalopathies[9].

The involvement of clinically or pathologically variable disease manifestations is also characteristic for a hereditary form of amyloidosis caused by transthyretin amyloid protein (ATTR). ATTR amyloidosis is one of the major forms of systemic amyloidosis, raising the question whether its variability may also be provoked by different fibril morphologies. Systemic ATTR amyloidosis depends on the misfolding of transthyretin (TTR)[10], which is natively a homo-tetrameric protein with a total mass of 55 kDa and 127 amino acid residues per protomer[11].

The general interest into this disease has greatly increased over the last decades, as systemic ATTR amyloidosis is aging-associated and as there are several disease-modifying pharmaceutical agents, which have become available to treat affected patients[12,13].

The spectrum of disease manifestations observed in different patients affected by this disease includes polyneuropathy, cardiomyopathy, renal failure along with several other symptoms[14,15]. There are early-onset and late-onset forms[16], and it is thought that the patient's phenotype is correlated to a certain allelic variant of TTR that is expressed in the respective patient[17]. Over 130 different mutational variants of the protein were found to be associated with systemic ATTR amyloidosis in addition to the wild type TTR (WT TTR)[18]. Many of these mutations were shown to destabilize the natively folded protein conformation such that they promote the dissociation and unfolding of tetramer[19,20]. But how these different mutations may lead to different disease manifestations and whether mutations and disease phenotype may be correlated with specific fibril morphologies is less well understood.

To investigate the relationship between mutation, disease manifestation and fibril morphology, we have determined the cryo-electron

[1]Institute of Protein Biochemistry, Ulm University, Helmholtzstrasse 8/1, Ulm D-89081, Germany. [2]Core Unit Mass Spectrometry and Proteomics, Medical Faculty, Ulm University, Ulm D-89081, Germany. [3]Medical Department V, Amyloidosis Center, Heidelberg, University Hospital Heidelberg, Im Neuenheimer Feld 400, Heidelberg D-69120, Germany. ✉e-mail: maximilian.steinebrei@uni-ulm.de

microscopy (cryo-EM) structures of several ATTR amyloid fibrils, which were isolated from the amyloidotic tissue of patients with hereditary ATTR amyloidosis. The patients expressed different mutations of TTR protein and showed specific clinical presentations. Based on cryo-EM, however, we find essentially the same fibril structure in all patients, demonstrating that the mutations do not necessarily lead to different fibril morphologies.

## Results

### Left-handed twist of isolated ATTR amyloid fibrils

ATTR amyloid fibrils were isolated from the explanted hearts of three patients that were diagnosed with systemic ATTR amyloidosis. Each patient was heterozygous for a different mutational variant of the TTR protein (V20I, G47E and V122I, mutational sites refer to mature TTR). Therefore, all three patients expressed WT TTR along with a different mutational variant of this protein. The patients varied also in the exact clinical and pathological disease manifestations, although all patients showed severe cardiomyopathy. The first patient (TTR V20I) is a man who was diagnosed at the age of 56. He underwent cardiac surgery one year later, due to acute heart failure, but no additional symptoms were noticed. The second patient is a woman (TTR G47E) who was diagnosed at the age of 56 years. She suffered from carpal tunnel syndrome and severe cardiomyopathy and cardiac surgery was performed six years later. The third patient is a man (TTR V122I) who was diagnosed at the age of 51 years. He also showed carpal tunnel syndrome and received a heart transplantation three years after diagnosis. None of the patients received anti-amyloid medication until heart transplantation.

The fibrils analyzed in this study were isolated from the explanted hearts with a previously established procedure to extract amyloid fibrils from patient tissue[21]. This method avoids harsh denaturing conditions and allowed us to obtain large quantities of elongated amyloid fibrils. TEM analysis of negatively stained and cryo-frozen fibril samples showed an almost monomorphic fibril distribution (Supplementary Fig. 1) and monomodal fibril width distribution centered at ~6.5 nm (Supplementary Fig. 2). The fibril cross-over structure was difficult to resolve with negatively stained TEM or cryo-EM images (Supplementary Fig. 1), similar to previous observations[22,23]. However,

scanning electron microscopy and platinum side shadowing demonstrated the left-hand twist of the isolated fibrils (Supplementary Fig. 1). Denaturing protein gel electrophoresis revealed the presence of several fibril protein bands that correspond to a molecular weight of 8–12 kDa (Supplementary Fig. 3). A similar pattern of protein bands was previously reported for the ATTR amyloid fibril proteins from WT and mutant TTR[22–24]. These data provide initial evidence that the three samples are dominated by a single fibril morphology, which is the same for the three patients.

### Cryo-EM structures of the isolated fibrils

Based on the recorded cryo-EM micrographs we reconstructed three-dimensional (3D) maps of the fibrils (Fig. 1, Supplementary Tables 1 and 2). For the three fibrils we obtained spatial resolutions of 3.18 Å (V20I) 2.37 Å (G47E) and 2.99 Å (V122I), based on the 0.143 Fourier shell correlation (FSC) criterion (Supplementary Fig. 4). During fibril picking, 2D or 3D classification, there was no evidence for the presence of significant levels of other fibril structures in our samples. That is, based on the visual inspection of the micrographs we did not observe any minor fibril populations, nor were particles or classes excluded after 2D classification. All maps were reconstructed with a left-hand twist, as suggested by platinum side shadowing (Supplementary Fig. 1). The 3D maps are very similar for the three fibrils (Fig. 1) and show almost identical helical parameters (Supplementary Table 1). The fibrils are polar, C1-symmetrical and share a highly similar, spearhead-like cross-sectional shape (Fig. 1). Each molecular layer consists of two discrete density regions that correspond to two ordered segments of the fibril protein: an N-terminal segment, extending from P11 to K35, and a C-terminal segment that spans from G57 to T123 of TTR protein (Fig. 1). Residues G1-C10, A36-H56 and N124-G127 are not seen in the 3D maps, although mass spectrometry (MS) showed their presence (see below). We conclude that the three segments are structurally disordered or cleaved off by proteolysis.

### The three fibrils share the same fibril protein fold

A superposition of the three structures shows that the fibril protein folds are virtually identical (Fig. 2a). The backbone root-mean-square deviation of all 92 resolved backbone Cα atoms was analyzed with

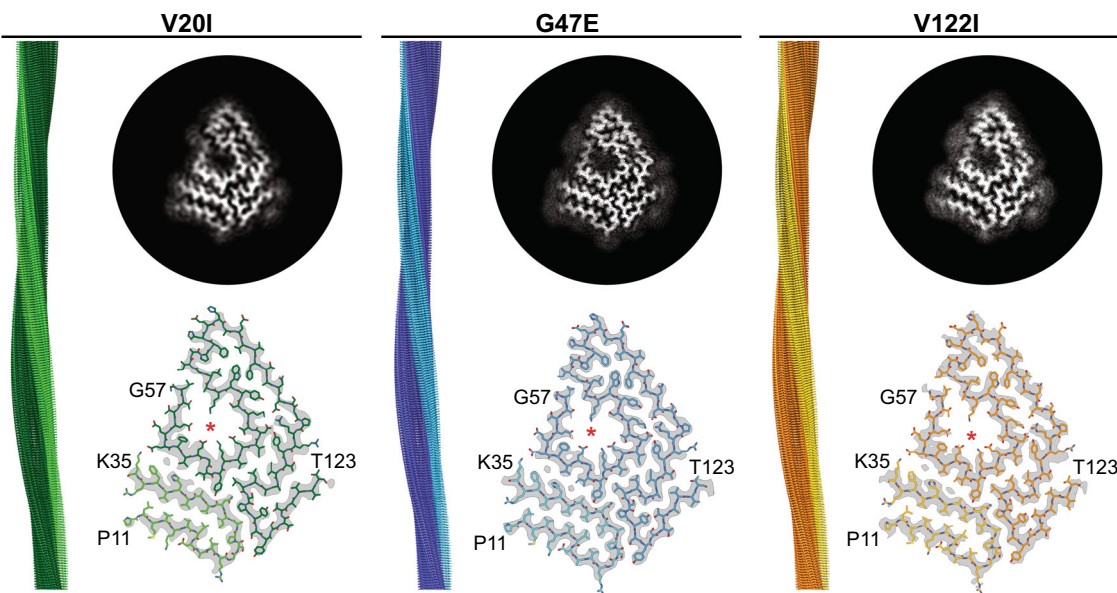

**Fig. 1 | 3D maps of ATTR fibrils from hereditary amyloidosis.** For each patient (V20I, G47E, V122I) the figure shows a side view of the reconstructed 3D map (left), a 5 Å thick fibril cross sectional slice of the 3D map (top right) and the molecular model superimposed with a one density layer slice of the map (gray, bottom right) are shown. The terminal amino acids are indicated, and the red asterisk marks the internal cavity. The color coding is consistent in the cross-sectional model and in the side view of the 3D map.

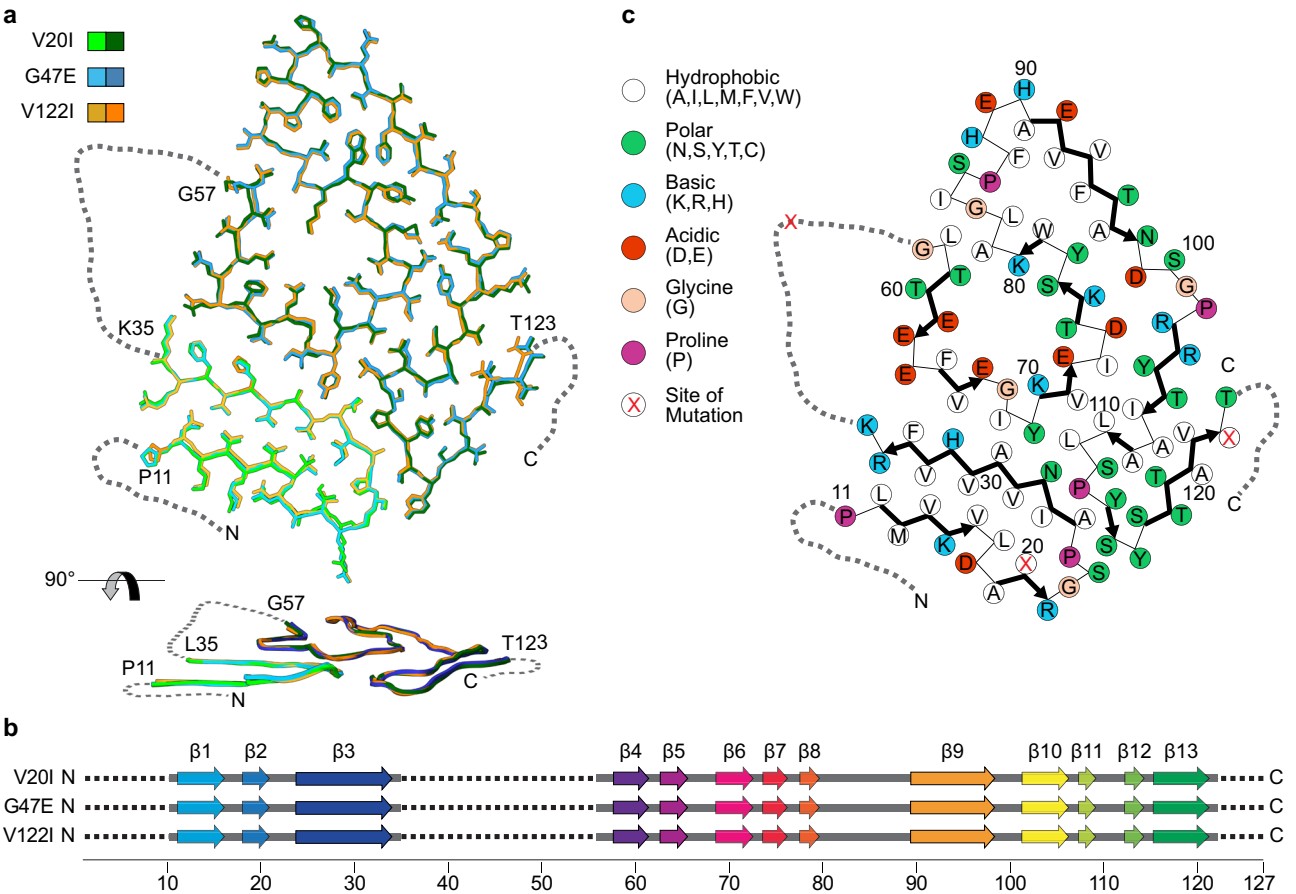

**Fig. 2 | Fold and β-sheet structure of the fibril proteins. a** Alignment of one molecular layer from each fibril structure. Top: cross sectional view; bottom: side view. The backbone root-mean-square deviation value ranges from 0.5 Å to 0.7 Å. **b** Schematic representation of the secondary structure of the three fibril proteins. Dotted lines represent the unresolved parts of the fibril proteins. **c** Schematic representation of the fibril protein fold. Zigzag line: polypeptide backbone; circles: amino acid residues; yellow circle: mutational sites; arrows: β-strands; dotted lines: unresolved parts of the 3D map.

Chimera[25] and found to range from 0.5 Å to 0.7 Å for the three structures. Each fibril protein contains thirteen β-strands that participate in the formation of thirteen cross-β sheets (Fig. 2b). The sheets present uniformly parallel strand-strand contacts in the direction of the fibril z-axis. Each fibril encloses a large internal cavity (Fig. 1) that is lined with polar and ionic residues (Fig. 2c), suggesting the presence of water. Each 3D map shows two small density features of uncertain molecular identity (Supplementary Fig. 5), which may reflect different possible conformations, for example of residue His31, or a molecular inclusion. As both density features have been reported previously for the amyloid fibril structures of a patient with SSA and a patient with V30M ATTR amyloidosis[22,23], they represent a general structural property of the investigated amyloid fibrils.

### Location of the mutation sites in the fibril structure

In a next step, we analyzed the position of the patient mutations in the three fibril structures. The site of the G47E mutation is not seen in the 3D map and located within the internal disordered segment (Fig. 3). The other two mutations (V20I and V122I) are seen in our 3D maps and occur within the fibril cross-β structure. Yet, they exist in different chemical environments: while residue 20 is part of the N-terminal fibril protein segment and located at a buried and hydrophobic position, residue 122 is located in the C-terminal fibril protein segment and affects a solvent-exposed position (Fig. 2c). Both mutational changes are chemically conservative and insert not more than a single methylene group into the mutant protein, all suggesting that the two mutations are not stabilizing or destabilizing to the fibril structure. This

conclusion is supported by the fact that the three fibril structures are essentially identical and the observation that the local resolution maps do not show any decrease of the resolution in the vicinity of the mutation site (Supplementary Fig. 4).

### The fibrils contain fragment from WT and mutant TTR protein

All three fibril samples contain both mutant and WT TTR protein (Supplementary Figs. 6–9, Supplementary Tables 3–5). This MS observation is consistent with the fact that the three patients heterozygously expressed the mutant TTR protein and demonstrate that the mutant protein is able to recruit the WT protein into the amyloid state. However, it is not known whether both mutant and WT protein exist within mixed fibrils or whether there are different fibril populations of fibrils that purely consist of WT and mutant TTR protein.

None of the fibrils contains detectable levels of full-length TTR protein. All observed fibril proteins are fragments of WT or mutant TTR protein (Supplementary Figs. 7–9). The proteolytic truncation occurred at different positions within sequence and fibril structure (Fig. 3). The most abundant proteolysis sites occur between residues F44 and S52 and thus within the internal disordered segment of the fibril protein. Several protein fragments are identical in the three analyzed fibril samples, such as the fragments P11-P43, F44-E127, A45-E127 and T49-E127 (Supplementary Figs. 6–9).

Some truncation sites occur within the ordered fibril core and are also conserved across the three different patients (Fig. 3a). These data show that positions from the well-structured part of the fibrils can be vulnerable to proteolysis. A proteolytic truncation at internal positions

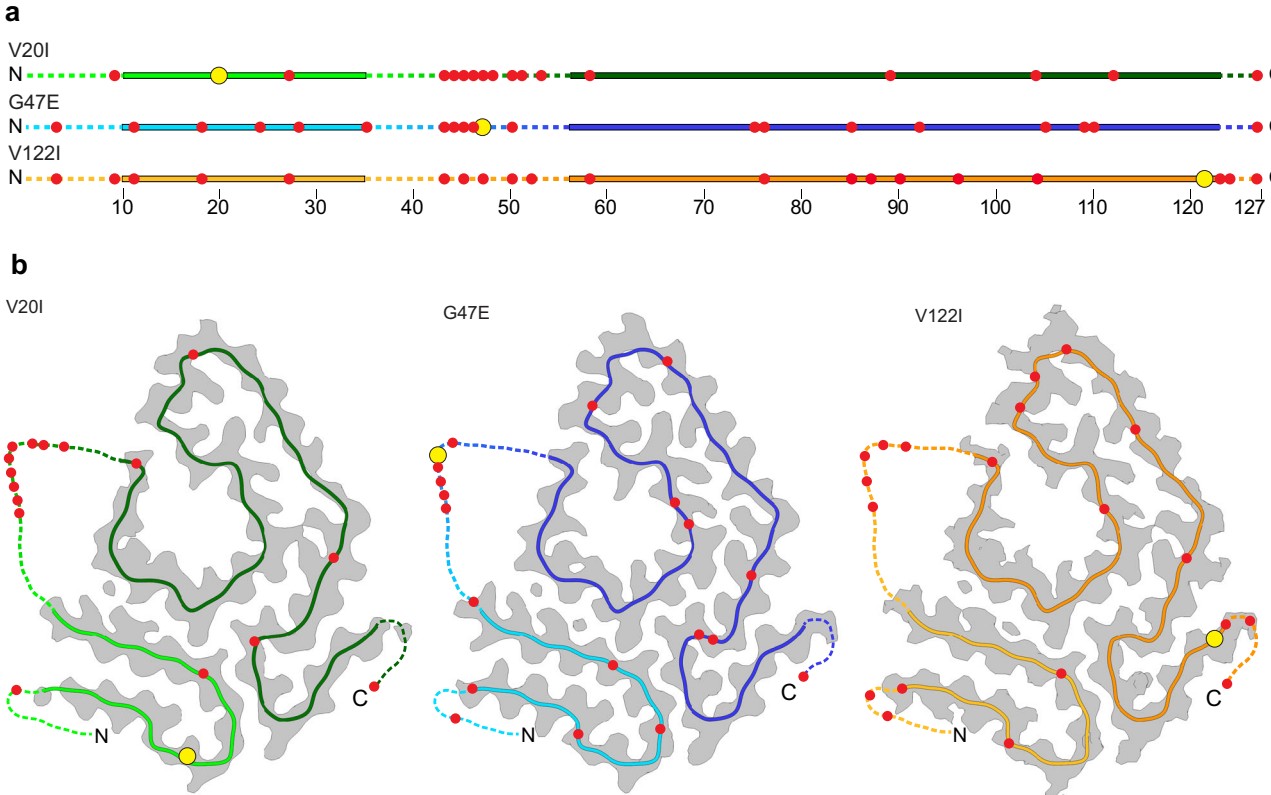

**Fig. 3 | Fragmentation sites of the fibril proteins. a** Red circles: location of the fragmentation sites in the sequence of the fibril proteins. Yellow circle: mutational sites, continuous lines: fibril core; dotted lines: TTR segments not seen in the 3D map. **b** Location of the fragmentation sites in the 3D structure as represented by a ribbon diagram of the molecular model (continuous line), which is superimposed with a section through the 3D map (gray). The yellow circle indicate the mutational sites. The dotted lines represent the unresolved part of TTR, drawn in arbitrary conformation. The figure is based the unambiguously assigned TTR fragments as outlined in Supplementary Tables 3–5.

of the fibril core structure has been reported for several other ex vivo amyloid fibril structures from systemic amyloidosis, including the amyloid fibril structure from systemic AA and AL amyloidosis[26–28]. The conservation of certain sites across different patients indicates a common mechanisms of fibril protein cleavage.

## Discussion

In this study we have analyzed the amyloid fibrils from several patients with hereditary ATTR amyloidosis. The fibrils are essentially monomorphic and structurally analogous in each patient. Moreover, they correspond to several previously reported ATTR amyloid fibril structures from other ATTR patients (Supplementary Fig. 10). These patients included individuals with senile systemic amyloidosis, who expressed WT TTR[23,29] as well as patients who expressed V30M[22,30], P24S and I84S[29] TTR. The close similarity of these fibril structures contrasts to the mutagenic and phenotypic variability of the respective patients and implies that the mutations underlying this disease do not necessarily lead to different fibril morphologies.

Instead, the main role of these mutations may be in the induction of the unfolding and dissociation of the native tetramer. That is, they promote misfolding by inducing the formation of relatively unfolded and highly aggregation-prone protein states. This native state-destabilizing effect was shown previously based on the biophysical analysis of proteins carrying disease-associated mutations, including the presently studied mutations V20I and V122I[19,20]. That the mutations affect the native state rather than the fibril structure was also concluded from a recent bioinformatic study of 36 mutations from hereditary ATTR amyloidosis which demonstrated that these mutations do not present a generally stabilizing or destabilizing effect on the

fibril structure[31]. In other words, there is no evidence that disease-associated mutations of TTR generally trigger amyloidosis by making the fibril state thermodynamically more favorable.

Yet, it is possible that patients with mutant TTR proteins show structurally altered fibrils. For example, it was found that some patients who express TTRV30M and TTRY114C show fibrils that consist mainly full length TTR protein. These fibrils differ from the fibrils that occur in the vast majority of ATTR patients, including the three presently studied mutations - and also other patients expressing TTRV30M or TTRY114C[24,32]. In case of patients expressing TTRV30M or TTRI84S, cryo-EM fibril structures were reported for some patients that differ from the canonical fibril structure described here (Supplementary Fig. 11). However, the differences affected only a relatively small part of the fibril protein fold (see below), while other patients with these two mutations did not show fibrils with the altered conformation[22,29]. In summary, altered amyloid fibril structures can be seen in some ATTR patients, while the majority of patients (also including other patients with the same mutation) contain fibrils that correspond to the one described here. We conclude that the respective mutations do not generally promote a different fibril structure and that patient-specific factors or circumstances are responsible for the formation of the altered fibril protein fold.

Furthermore, we noted that the structural variability of fibrils from V30M or I84S patients affects mainly residues G57-Y69 (Supplementary Fig. 11), which occur immediately after the C-terminal end of the internal disordered segment (Fig. 4). The internal disordered segment is not seen in the 3D map, and it is partially cleaved off in the fibril, as demonstrated by MS (Fig. 3a). The conformational heterogeneity of the fibril protein adjacent to the internal disordered

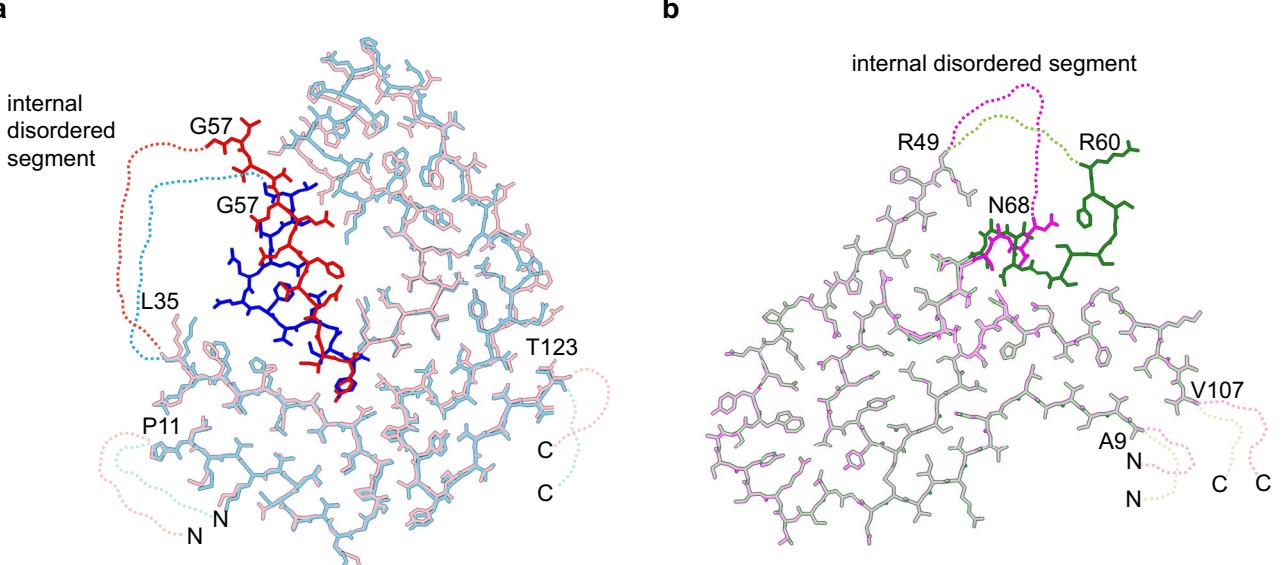

**Fig. 4 | Fibril protein heterogeneity adjacent to the internal disordered segment. a** Systemic ATTR amyloidosis. Overlay of stick representation of ATTR V30M amyloid fibrils from two patients (PBD: 6SDZ[22] and 7OB4[30]). **b** Systemic AL amyloidosis. Overlay of two lambda light chain-derived amyloid fibril structures from patient FOR005 (PDB: 6Z1O and 6Z1I)[26].

segment therefore suggests that disordered segments may affect the structure of adjacent residues in the fibril core, at least in some patients and fibrils. A similar case was reported recently for the light chain-derived amyloid fibrils from patient FOR005, who suffered from systemic AL amyloidosis[26]. This patient also showed conformational heterogeneity in a small segment of the ordered fibril core (residues R49-N68) that occurred immediately C-terminal to an internal disordered segment (Fig. 4).

So far, it has been difficult to reconstitute fibril structures in vitro that match the structural properties of ex vivo fibrils[13,33,34]. While there is evidence for TTR aggregation in the mildly acidic range around pH 4, the formed aggregates formed did not show the typical, linear structure of the ex vivo fibrils described here[33]. By contrast, much better defined fibrils were obtained in vitro with fragmented TTR protein, and it was thus suggested that TTR fragmentation could be crucial for disease biogenesis in vivo[35,36]. However, ex vivo ATTR amyloid fibrils do contain N- and C-terminal segments of TTR, and the relative arrangement of the two strongly argued that that the fibrils were actually formed from full-length protein and became cleaved after fibril formation[22]. Indeed, a recent in vitro seeding study with patient-derived ATTR amyloid fibrils demonstrated the formation of bona fide amyloid fibrils structures within the test tube[37]. Hence, TTR is able to assemble into well-defined fibril structures in vitro, if the sample contains appropriate fibril seeds. While it is so far unknown what factors might determine the formation of this nucleus inside the body, it is likely that this reaction is influenced by cellular factors, such as lipids, glycosaminoglycans or proteases. And indeed, proteolytic resistance recently emerged as a common theme in ex vivo amyloid fibrils, suggesting that this property enabled specific fibril morphologies to accumulate and to cause problems in vivo[38].

Taken together, the spectrum of mutation-dependent disease manifestations in systemic ATTR amyloidosis does not necessarily arise from the formation of different fibril morphologies. Instead, it may be provoked, at least in the majority of cases, by the circumstances under which the respective fibril structures are formed and deposited within the tissue. A mutation may influence the effect of these circumstances as it could make unfolding more favorable, such that the accumulation of fibril starts at a relatively early time point in life. Alternatively, it may influence the interactions of the tetramer,

unfolded TTR states or early aggregates with certain tissue factors or cells, which leads to a certain deposition pattern of amyloid fibrils in the tissue or throughout the body. Ultimately, it could lead to the stabilization of certain oligomeric aggregate structures, which are not fibrils but nevertheless harmful to the surrounding environment. While further work is required to establish the mechanisms through which mutations in TTR are responsible for causing the variable phenotypic manifestations of this disease, our data demonstrates that this variability does not solely arise from the formation of different fibril morphologies.

## Methods

### Patient description

This study is based on analyses of cardiac tissue from explanted hearts obtained from three patients with hereditary ATTR amyloidosis. For each patient, the four exons of the transthyretin gene were sequenced from blood samples to identify the genetic variants. Samples from patients ATTRV20I and ATTRG47E were used in a previous study[39]. Following the recommendations by the International Society of Amyloidosis[40], the amino acid sequences in this study refer to the sequence of the mature TTR protein after removal of the signal sequence. The study was approved by the ethical committees of the University of Heidelberg (S-123/2006) and of Ulm University (103/21). Informed consent for publication of patient details was obtained from the patient, as well as consent for the use of these samples for scientific research.

### Fibril extraction

Amyloid fibrils were extracted from the collected tissue, according to a previously described extraction protocol[21]. In brief, 125 mg frozen human heart tissue were diced with a scalpel and washed with 500 μL ice cold tris hydroxymethyl aminomethane (Tris)-calcium buffer [20 mM Tris, 138 mM NaCl, 2 mM CaCl$_2$, 0.1% (w/v) NaN$_3$, pH 8.0]. The sample was centrifuged for 5 min at 3100 × $g$ and 4 °C. The supernatant was removed, and the pellet was resuspended in 500 μL ice-cold Tris-calcium buffer. The suspension was homogenized with a Kontes Pellet Pestle and centrifuged again for 5 min. This Tris-calcium buffer washing step was repeated four more times. The pellet of the last centrifuge step was resuspended in 1 mL freshly prepared 5 mg/mL *Clostridium*

*histolyticum* collagenase (Sigma) in Tris-calcium buffer with ethylene-diaminetetraacetic acid (EDTA)-free protease inhibitor (Roche). The solution was incubated overnight on a shaker at 37 °C and centrifuged for 30 min at 3100 × *g* and 4 °C. The supernatant was removed, and the pellet was resuspended in 500 μL Tris-EDTA buffer [20 mM Tris, 140 mM NaCl, 10 mM EDTA, 0.1% (w/v) NaN₃, pH 8.0] and centrifuged for 5 min at 3100 × *g* and 4 °C. This Tris-EDTA buffer washing step was repeated two more times. The pellet was finally resuspended in 100 μL ice-cold water and centrifuged for 5 min at 3100 × *g* at 4 °C. The fibril-containing supernatant was retained. The water extraction step was repeated five more times to generate six supernatant fractions.

### Mass spectrometry

A 500 μL aliquot of each fibril solution was lyophilized and resuspended in 6 M guanidine hydrochloride, buffered with 50 mM Tris-HCl and pH 8 to reach a protein concentration of 0.5 mg/ml. After an overnight incubation step, an aliquot containing 1 μg of protein was adjusted to 15 μL using 0.1% (v/v) trifluoroacetic acid (TFA), and was applied onto a U3000 RSLCnano (Thermo Fisher Scientific) column. The eluate was applied online onto an LTQ Orbitrap Elite system (Thermo Fisher Scientific) that was equipped with a nano-electrospray ion source and distal coated SilicaTips (FS360-20-10-D, New Objective). The samples were analyzed in the positive ion mode using a spray voltage of 1.5 kV. The mass spectra were acquired from 370 to 1700 m/z in the ion trap of the instrument using a normal scan mode. The resolution was set to 30,000 (at 400 m/z). The automatic gain control was enabled and set to $10^6$ ions with a maximum fill time of 500 ms. The raw data was deconvoluted by the MASH Explorer[41] using default settings and the "Quick Deconvolution" feature. To avoid artefacts, the deconvoluted spectra contain only peaks, where the monoisotopic masses could be assigned with confidence score of at least 90% and which resulted from m/z peaks with 5 charge or more. Masses were manually assigned to the WT and variant TTR sequences using the software mMass v5.5.0[42].

### Electron microscopy of negatively stained samples

To prepare the analysis specimens formvar/Carbon 200 mesh copper grids (Electron Microscopy Sciences) were glow-discharged at 20 mA for 40 s with a PELCO easiGlow instrument (TED PELLA). A 3.5 μL aliquot of the fibril solution was placed onto the grid. After incubation of the sample for 45 s, excess solvent was blotted away with filter paper (Whatman). The grids were stained three times with 10 μL of a 2% (w/v) uranyl acetate solution. The grids were imaged with a JEM-1400 TEM (Joel) that was operated at 120 kV and equipped with a F216 camera (TVIPS).

### Platinum side shadowing and scanning electron microscopy

Formvar/Carbon 200 mesh copper grids (Electron Microscopy Sciences) were glow-discharged for 40 s using a PELCO easiGlow™ glow discharge system operated at 20 mA. A 3.5 μL aliquot of the fibril sample was applied onto the grid and excess solvent was blotted away with filter paper. The grids were dried at room temperature. Using a Blazers TKR 010 instrument, a 1 nm thick layer of platinum was evaporated at an angle of 30°. The grids were analyzed using a Hitachi S-5200 scanning electron microscope (Hitachi) at 10 kV acceleration voltage.

### Cryo-EM sample preparation and data collection

C-flat 1.2/1.3 400 mesh holey carbon cupper grids (Electron Microscopy Sciences) were glow-discharged for 40 s using a PELCO easiGlow™ glow discharge system operated at 20 mA. The grids were mounted in an EM GP2 (Leica) and a 3.5 μL aliquot of the fibril solution was applied before they were blotted with filter paper for 4 s at 21 °C and 90% relative humidity. The grids were plunge frozen in liquid ethane and controlled, if necessary, with a 2100 F transmission electron microscope (Joel). Details of the cryo-EM data collection is listed in Supplementary Table 1. The fibril width was measured on the micrographs using Fiji (Image j)[43].

### Reconstruction of the 3D maps

The raw data move frames were gain-corrected using IMOD[44] and in case of the falcon 4 data, the movie frames were summed into a single micrograph using the RELION 3.1 EER implementation[45]. Motion correction and dose-weighting was done using MOTIONCOR 2.1[46]. The contrast transfer function was estimated from the motion-corrected images using CTFFIND-4.1[47]. All subsequent image-processing steps were performed using the helical reconstruction methods implemented in RELION 3.1[48,49]. The fibrils were picked manually and extracted with a box size of 256 pixel and an inter box distance of 10%. A reference free 2D classification was performed to remove low-quality segments. The remaining segments were subjected to a 3D classification using a featureless cylinder as the starting reference structure. A combination of 3D classification and 3D auto refinement steps were carried out to select the best segments. contrast transfer function -refinement[50] and Bayesian polishing[51] were performed to further increase the resolution. See Supplementary Table 1 for further details.

### Model building and refinement

The previously described structure of the ATTRwt amyloid fibril, protein data bank (PDB) entry 8ADE, was used as a starting structure and altered, if necessary, at the position of mutation using Chimera[25]. Each model was built individually. The manual refinement was done using Coot[52] with non-crystallographic symmetry constraints, Ramachandran, atomic displacement parameter and rotamer restraints. The atomic clashes, rotamer and Ramachandran outliers and model geometry were analyzed by the validation output generated using MolProbity[53] and the comprehensive validation tool in Phenix[54]. Once a satisfactory main and side-chain density fit was achieved for one polypeptide chain, a fibril stack comprising six poly-peptide chains was assembled using the pdbsymm tool implemented in Situs[55]. The described cycle of iterative refinement and modeling was repeated for the fibril stack until a reasonable density to model fit was achieved. Supplementary Table 2 gives more details on the modeling.

### Reporting summary

Further information on research design is available in the Nature Portfolio Reporting Summary linked to this article.

## Data availability

The reconstructed cryo-EM maps were deposited in the Electron Microscopy Data Bank with accession codes EMD-17736 (ATTRV20I), EMD-17737 (ATTRG37E) and EMD-17738 (ATTRV122I). The coordinates of the fitted atomic models were deposited at the PDB under the accession codes 8PKE (ATTRV20I), 8PKF (ATTRG37E) and 8PKG (ATTRV122I). The cryo-EM data were deposited on EMPIAR with the accession code EMPIAR-11700 (V20I), EMPIAR-11730 (G47E) and EMPIAR-11704 (V122I). Source data are provided with this paper for the following figures: Supplementary Fig. 1, Supplementary Fig. 2, Supplementary Fig. 3, Supplementary Fig. 4 and Supplementary Fig. 5. Source data are provided with this paper.

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

## Acknowledgements

This project was supported by the Deutsche Forschungsgemeinschaft (grant no. FA 456/28 to M.F). We thank Natalie Scheuermann and Paul Walther (Ulm University) as well as Felix Weis (European Molecular Biology Laboratory, Heidelberg) for technical support. The collection of the cryo-EM datasets was supported by iNEXT Discovery (17206 and 17128), which were collected at the European Molecular Biology Laboratory, Heidelberg.

## Author contributions

M.St., J.B., A.P., N.K. and S.W. performed research. U.H., and S.O.S. contributed materials. M.St., J.B., and M.Sch. analyzed the data. M.St., M.Sch., and M.F. wrote the paper with contributions from all other authors.

## Funding

## Competing interests

The authors declare no competing interest.
