## [Peer review file · Nature Communications]

REVIEWER COMMENTS

Reviewer #1 (Remarks to the Author):

The manuscript "Common transthyretin-derived amyloid fibril structures in patients with hereditary ATTR amyloidosis" by Steinebrei et al describes the ex vivo structures of the amyloid deposits found in three ATTR patients carrying three known amyloidogenic TTR mutations (V20I, G47E and V122I). The manuscript is clear, well written and reports solid and relevant structural data. I recommend this manuscript for publication with minor revisions.

Here some comments and suggestions follow:

Line 113: was there only one 2D class for each of the datasets? Were all particles included in the 2D class shown in the manuscript? Or is there any evidence of minor structural population? Please discuss this more clearly in the text.

Line 131: state the number of Ca used for the rmsd calculations (all Ca I guess...)

Line 176: in this context it could be a good idea adding two references (10.1111/febs.1618, 10.1074/jbc.RA120.013461) dealing with proteolysis of ex vivo fibrils in AL amyloidosis.

Line 193: could the G47E mutation have the same effect? I.e. no contribution for the fibril stabilisation but would be possible to speculate about a destabilisation of the native state?

Discussion: The authors make the reasonable claim that the three mutations they have analysed in this paper do not have any significant impact on the fibrillar structure. Moreover, the authors seem to suggest that other TTR mutations may affect mostly the native state and not the fibrillar state. In this direction it would be interesting to analyse which of the known TTR amyloidogenic mutations are structurally/biochemically compatible/incompatible with the fibrillar fold seen in these structures and in the previous ones. This will not allow concluding that all TTR amyloids will/will not display the same structure but having a significant number of structures now available, I think it is a timely analysis to perform.

Reviewer #2 (Remarks to the Author):

This is an important study investigating amyloid polymorphism in ATTR amyloidosis. The manuscript is clearly written and the text and figures are easy to follow. The study methods are appropriate and the author's conclusions are strongly supported by their data. Findings are novel and will be of considerable interest to researchers in ATTR amyloidosis and more widely in the protein misfolding community.

I have only three minor corrections for the authors to consider.

1. Line 51. ATTR should be defined at first occurrence, as amyloid transthyretin (ATTR)
2. Line 195, the word of should be changed to or (to read, ...stabilising or destabilising...).
3. Lines 218-219. In this sentence reference to Figure 3a was confusing as residues G57-Y69 are part of the core in the three fibrils shown. Please can the authors review their wording?

Reviewer #3 (Remarks to the Author):

This is a potentially important paper suggesting that unique amyloid fibril morphology can result in the phenotypic variability in patients. A contrasting concept is that different fibril morphologies are linked to distinct pathological and clinical manifestations. When the authors studies fibrils structures of three

TTR mutants with cryo-EM microscopy, the atomic structures were remarkably similar, suggesting that the mutation-dependent disease manifestations can be caused by unknown circumstantial factors. Although the paper is interesting, I have some concerns regarding novelty of the paper.

1. The authors previously reported the cry-EM structure of V30M TTR amyloid fibrils (Schmidt et al. *Nat. Commun.*, 2019), which seems very similar to those reported here. The similar structure of V30M TTR amyloid fibrils was also reported by Iakovleva et al. (*Nat. Commun.*, 2021) and the two structures were compared and discussed in Discussion (Fig. 4). Then, what is novelty? If this is the first paper proposing the one amyloid structure-varying phenotypes, all four (or five) TTR amyloid structures should be compared all together in this paper to emphasize the common fibril morphology.
2. The role of TTR fragmentation in amyloid formation seems complicated and still unclear. In the previous paper (Schmidt et al. *Nat. Commun.*, 2019), they included a schematic figure with early fibril state, which was helpful (Fig. 6). The authors should include another schematic figure with their advanced idea of the role of TTR fragmentation.
3. Line 204-: "In case of V30M and Y114C patients, it was found that some patients show fibrils that consist mainly full length TTR protein, while —". To my understanding, nobody succeeded in in vitro amyloid formation with full length TTR, suggesting that fragmentation is essential, at least in vitro, to form mature amyloid fibrils. The authors may address the difference between in vivo and in vitro mechanisms of TTR amyloid fibril formation.

REVIEWER COMMENTS

Reviewer #1

The manuscript “Common transthyretin-derived amyloid fibril structures in patients with hereditary ATTR amyloidosis” by Steinebrei et al describes the ex vivo structures of the amyloid deposits found in three ATTR patients carrying three known amyloidogenic TTR mutations (V20I, G47E and V122I). The manuscript is clear, well written and reports solid and relevant structural data. I recommend this manuscript for publication with minor revisions. Here some comments and suggestions follow:

1.

Line 113: was there only one 2D class for each of the datasets? Were all particles included in the 2D class shown in the manuscript? Or is there any evidence of minor structural population? Please discuss this more clearly in the text.

Response:

We thank this referee for providing these very positive and supportive comments and for taking the time to evaluate our manuscript. 2D classifications were performed separately for each data set. They resulted in several 2D classes per data set and all classes seemed to belong to one fibril morphology. No particles were excluded at the 2D classification stage. We noted that this was ambiguous in our previous description as the Table Section “Number of segments after 2D classification” was missing. For clarity, we now added this information to Supplementary Table 1. We did not obtain 2D or 3D classes that suggested the presence of a minor morphology. This is now expressed in the revised manuscript as: “That is, based on the visual inspection of the micrographs we did not observe any minor fibril populations. Nor were particles or classes excluded after 2D classification”. (Lines 118-120)

2.

Line 131: state the number of Ca used for the rmsd calculations (all Ca I guess...)

Response:

Thank you for this comment. We calculated the RMSD based on all 92 C α atoms of the molecular model that was fitted to the 3D map. This is now expressed in the manuscript as follows: “The backbone root-mean-square deviation of all 92 resolved backbone C α atoms was analyzed with Chimera and found to range from 0.5 Å to 0.7 Å for the three structures”. (Lines 133-134)

3.

Line 176: in this contest it could be a good idea adding tow references (10.1111/febs.1618, 10.1074/jbc.RA120.013461) dealing with proteolysis of ex vivo fibrils in AL amyloidosis.

Response:

Thank you for this suggestion. We now added one of suggested references, as it indeed shows similar features in AL amyloidosis as we report here for ATTR amyloidosis. (Line 180)

4.

Line 193: could the G47E mutation have the same effect? I.e. no contribution for the fibril stabilisation but would be possible to speculate about a destabilisation of the native state?

Response:

We did not mention G47E in this context as we could not find experimental studies demonstrating such an effect for the G47E mutation. However, we assume that G47E might also be destabilizing because position 47 is located within the CD loop of TTR, which is

generally considered as crucial for the stability of the native protein; and replacing a glycine residue with glutamic acid may well influence the properties of this loop.

5.

Discussion: The authors make the reasonable claim that the three mutations they have analysed in this paper do not have any significant impact on the fibrillar structure. Moreover, the authors seem to suggest that other TTR mutations may affect mostly the native state and not the fibrillar state. In this direction it would be interesting to analyse which of the known TTR amyloidogenic mutations are structurally/biochemically compatible/incompatible with the fibrillar fold seen in these structures and in the previous ones. This will not allow concluding that all TTR amyloids will/will not display the same structure but having a significant number of structures now available, I think it is a timely analysis to perform.

Response:

Thank you for this suggestion, but it seems that a similar analysis was already performed by another group, at least with a set of 36 mutations from hereditary ATTR amyloidosis (DOI 10.1002/prot.26399). Although we cited this study in our discussion it is possible that its content did not become clear from our previous description. Therefore, we have changed the respective sentences so that it now reads as: "That the mutations affect the native state rather than the fibril structure was also concluded from a recent bioinformatic study of 36 mutations from hereditary ATTR amyloidosis which demonstrated that these mutations do not present a generally stabilizing or destabilizing effect on the fibril structure. In other words, there is no evidence that disease-associated mutations of TTR generally trigger amyloidosis by making the fibril state thermodynamically more favorable". (Lines 200-205)

Reviewer #2

This is an important study investigating amyloid polymorphism in ATTR amyloidosis. The manuscript is clearly written and the text and figures are easy to follow. The study methods are appropriate and the author's conclusions are strongly supported by their data. Findings are novel and will be of considerable interest to researchers in ATTR amyloidosis and more widely in the protein misfolding community. I have only three minor corrections for the authors to consider.

1.

Line 51. ATTR should be defined at first occurrence, as amyloid transthyretin (ATTR)

Response:

We thank this referee for the positive feedback which helps to make our paper stronger. Following the latest recommendations of the Nomenclature Committee of ISA (10.1080/13506129.2022.2147636) we now define ATTR as: "transthyretin amyloid protein (ATTR)". (Lines 50-53)

2.

Line 195, the word of should be changed to or (to read, ... stabilizing or destabilizing ...).

Response:

Thank you very much for spotting this mistake. We corrected the wrong sentence. (Line 202)

3.

Lines 218-219. In this sentence reference to Figure 3a was confusing as residues G57-Y69 are part of the core in the three fibrils shown. Please can the authors review their wording?

Response:

Thank you for this remark. Indeed, our previous sentence was confusing and we changed it accordingly so that it now reads as: “Furthermore, we noted that the structural variability of fibrils from V30M or I84S patients affects mainly residues G57-Y69 (Supplementary Figure 11), which occur immediately after the C-terminal end of the internal disordered segment (Figure 4). (Lines 222-224)

Reviewer #3

This is a potentially important paper suggesting that unique amyloid fibril morphology can result in the phenotypic variability in patients. A contrasting concept is that different fibril morphologies are linked to distinct pathological and clinical manifestations. When the authors studies fibrils structures of three TTR mutants with cryo-EM microscopy, the atomic structures were remarkably similar, suggesting that the mutation-dependent disease manifestations can be caused by unknown circumstantial factors. Although the paper is interesting, I have some concerns regarding novelty of the paper.

1.

The authors previously reported the cry-EM structure of V30M TTR amyloid fibrils (Schmidt et al. Nat. Commun., 2019), which seems very similar to those reported here. The similar structure of V30M TTR amyloid fibrils was also reported by Iakovleva et al. (Nat. Commun., 2021) and the two structures were compared and discussed in Discussion (Fig. 4). Then, what is novelty? If this is the first paper proposing the one amyloid structure-varying phenotypes, all four (or five) TTR amyloid structures should be compared all together in this paper to emphasize the common fibril morphology.

Response:

We thank this referee for critically reading our manuscript and raising these concerns. This will give us the opportunity to address them. We have now added a new figure to our manuscript (Supplementary Figure 10), which shows all currently accessible 3D maps of ATTR amyloid fibrils. We show the 3D maps because for several patients there are only the 3D maps accessible (but no PDB files). This will give the best possible overview on the now available cryo-EM data from different patients.

2.

The role of TTR fragmentation in amyloid formation seems complicated and still unclear. In the previous paper (Schmidt et al. Nat. Commun., 2019), they included a schematic figure with early fibril state, which was helpful (Fig. 6). The authors should include another schematic figure with their advanced idea of the role of TTR fragmentation.

Response:

To address this and the following comment we added a new paragraph to our discussion and a new figure to the SI (Supplementary Figure 11). According to this scheme, it is possible that factors such as proteolytic fragmentation affect the initial nucleation step and the formation of an appropriate fibril nucleus.

3.

Line 204-: “In case of V30M and Y114C patients, it was found that some patients show fibrils that consist mainly full length TTR protein, while —“. To my understanding, nobody succeeded in in vitro amyloid formation with full length TTR, suggesting that fragmentation is essential, at least in vitro, to form mature amyloid fibrils. The authors may address the difference between in vivo and in vitro mechanisms of TTR amyloid fibril formation.

Response:

This is now addressed in a new section of our discussion. In brief, we agree to the referee that the de novo formed fibrils structures are, in in vitro experiments, often not very linear and morphologically different from the ex vivo fibril structures. However, using seeding with ex vivo fibrils, there are indeed reports of the formation of appreciably linear fibrils - also in vitro (10.1073/pnas.1805131115). Therefore, it is also possible to form well-resolved TTR fibrils in vitro, provided that the correct seeds are present. This narrows the discussion to the first step, the de novo nucleation reaction, which may well be affected by proteolysis (or other factors of the native cellular environment).